# Learning Functionally Decomposed Hierarchies for Continuous Navigation Tasks

## Abstract

Solving long-horizon sequential decision making tasks in environments with sparse rewards is a longstanding problem in reinforcement learning (RL) research. Hierarchical Reinforcement Learning (HRL) has held the promise to enhance the capabilities of RL agents via operation on different levels of temporal abstraction. Despite the success of recent works in dealing with inherent nonstationarity and sample complexity, it remains difficult to generalize to unseen environments and to transfer different layers of the policy to other agents. In this paper, we propose a novel HRL architecture, *Hierarchical Decompositional Reinforcement Learning* (HiDe), which allows decomposition of the hierarchical layers into independent subtasks, yet allows for joint training of all layers in end-to-end manner. The main insight is to combine a control policy on a lower level with an image-based planning policy on a higher level. We evaluate our method on various complex continuous control tasks for navigation, demonstrating that generalization across environments and transfer of higher level policies can be achieved. See videos https://sites.google.com/view/hide-rl.

## 1 Introduction

Reinforcement learning (RL) has been succesfully applied to sequential-decision making tasks, such as learning how to play video games in Atari (Mnih et al., 2013), mastering the game of Go (Silver et al., 2017) or continuous control in robotics (Lillicrap et al., 2015; Levine et al., 2015; Schulman et al., 2017). However, despite the success of RL agents in learning control policies for myopic tasks, such as reaching a nearby target, they lack the ability to effectively reason over extended horizons. In this paper, we consider continuous control tasks that require planning over long horizons in navigation environments with sparse rewards. The task becomes particularly challenging with sparse and delayed rewards since an agent needs to infer which actions caused the reward in a domain where most samples give no signal at all. Common techniques to mitigate the issue of sparse rewards include learning from demonstrations (Schaal, 1999; Peng et al., 2018) or using enhanced exploration strategies (Bellemare et al., 2016; Pathak et al., 2017; Andrychowicz et al., 2017).

Hierarchical Reinforcement Learning (HRL) has been proposed in part to solve such tasks. Typically, a sequential decision making task is split into several simpler subtasks of different temporal and functional abstraction levels (Sutton et al., 1999; Andre & Russell, 2002). Although the hierarchies would ideally be learned in parallel, most methods resort to curriculum learning (Frans et al., 2017; Florensa et al., 2017; Bacon et al., 2016; Vezhnevets et al., 2017). Recent goal-conditioned hierarchical architectures have successfully trained policies jointly via off-policy learning (Levy et al., 2019; Nachum et al., 2018; 2019). However, these methods often do not generalize to unseen environments as we show in Section 5.1. We argue that this is due to a lack of true separation of planning and low-level control across the hierarchy. In this paper, we consider two main problems, namely functional decomposition of HRL architectures in navigation-based domains and generalization of RL agents to unseen environments (figure 1).

To address these issues, we propose a novel multi-level HRL architecture that enables both *functional decomposition* and *temporal abstraction*. We introduce a 3-level hierarchy that decouples the major roles in a complex navigation task, namely planning and low-level control. The benefit of a modular design is twofold. First, layers have access to only task-relevant information for a predefined task, which significantly improves the generalization ability of the overall policy. Hence, this enables policies learned on a single task to solve randomly configured environments. Second,

Training                    Test

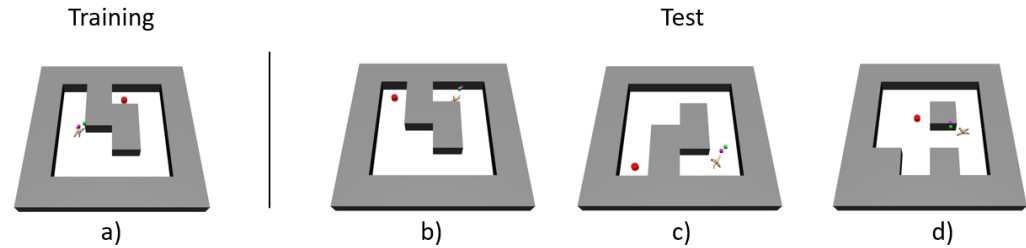

      a)                b)                c)                d)

Figure 1: Navigation environments. The red sphere indicates the goal an agent needs to reach, with the starting point at the opposite end of the maze. The agent is trained on environment a). To test generalization, we use the environments with b) reversed starting and goal positions, c) mirrored maze with reversed starting and goal positions and d) randomly generated mazes.

the planning and control layers are modular and thus allow for composition of cross-agent architectures. We empirically show that the planning layer of the hierarchy can be transferred successfully to new agents. During training we provide global environment information only to the planning layer, whereas the full internal state of the agent is only accessible by the control layer. The actions of the top and middle layers are in the form of displacement in space. Similarly, the goals of the middle and lowest layers are relative to the current position. This prevents the policies from overfitting to the global position in an environment and hence encourages generalization to new environments.

In our framework (see figure 2), the planner (i.e., the highest level policy $\pi_2$) learns to find a trajectory leading the agent to the goal. Specifically, we learn a value map of the environment by means of a value propagation network (Nardelli et al., 2019). To prevent the policy from issuing too ambitious subgoals, an attention network estimates the range of the lower level policy $\pi_0$ (i.e., the agent). This attention mask also ensures that the planning considers the agent performance. The action of $\pi_2$ is the position which maximizes the masked value map, which serves as goal input to the policy $\pi_1$. The middle layer implements an interface between the upper planner and lower control layer, which refines the coarser subgoals into shorter and reachable targets for the agent. The middle layer is crucial in functionally decoupling the abstract task of planning ($\pi_2$) from agent specific continuous control. The lowest layer learns a control policy $\pi_0$ to steer the agent to intermediate goals. While the policies are functionally decoupled, they are trained together and must learn to cooperate.

In this work, we focus on solving long-horizon tasks with sparse rewards in complex continuous navigation domains. We first show in a maze environment that generalization causes challenges for state-of-the-art approaches. We then demonstrate that training with the same environment configuration (i.e., fixed start and goal positions) can generalize to randomly configured environments. Lastly, we show the benefits of functional decomposition via transfer of individual layers between different agents. In particular, we train our method with a simple 2DoF ball agent in a maze environment to learn the planning layer which is later used to steer a more complex agent. The results indicate that the proposed decomposition of policy layers is effective and can generalize to unseen environments. In summary our main contributions include:

- A novel multi-layer HRL architecture that allows functional decomposition and temporal abstraction for navigation tasks.
- This architecture enables generalization beyond training conditions and environments.
- Functional decomposition that allows transfer of individual layers across different agents.

## 2 RELATED WORK

Learning hierarchical policies has seen lasting interest (Sutton et al., 1999; Schmidhuber, 1991; Dietterich, 1999; Parr & Russell, 1998; McGovern & Barto, 2001; Dayan & Hinton, 2000), but many approaches are limited to discrete domains or induce priors.

More recent works (Bacon et al., 2016; Vezhnevets et al., 2017; Tirumala et al., 2019; Nachum et al., 2018; Levy et al., 2019) have demonstrated HRL architectures in continuous domains. Vezhnevets et al. (2017) introduce FeUdal Networks (FUN), which was inspired by feudal reinforcement learn-

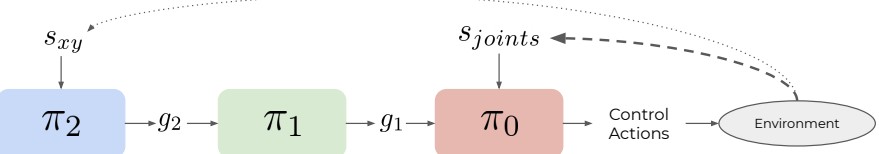

Figure 2: Our 3-layer HRL architecture. The planning layer $\pi_2$ receives a birds eye view of the environment and the agent's position $s_{xy}$ and sets an intermediate target position $g_2$. The interface layer $\pi_2$ splits this subgoal into reachable targets $g_1$. A goal-conditioned control policy $\pi_0$ learns the required motor skills to reach the target $g_1$ given the agent's joint information $s_{joints}$.

ing (Dayan & Hinton, 2000). In FUN, a hierarchic decomposition is achieved via a learned state representation in latent space. While being able to operate in continuous state space, the approach is limited to discrete action spaces. Tirumala et al. (2019) introduce hierarchical structure into KL-divergence regularized RL using latent variables and induces semantically meaningful representations. The separation of concerns between high-level and low-level policy is guided by information asymmetry theory. Transfer of resulting structure can solve or speed up training of new tasks or different agents. Nachum et al. (2018) present HIRO, an off-policy HRL method with two levels of hierarchy. The non-stationary signal of the upper policy is mitigated via off-policy corrections, while the lower control policy benefits from densely shaped rewards. Nachum et al. (2019) propose an extension of HIRO, which we call HIRO-LR, by learning a representation space from environment images, replacing the state and subgoal space with neural representations. While HIRO-LR can generalize to a flipped environment, it needs to be retrained, as only the learned space representation generalizes. Contrarily, HiDe generalizes without retraining. Levy et al. (2019) introduce Hierarchical Actor-Critic (HAC), an approach that can jointly learn multiple policies in parallel. The policies are trained in sparse reward environments via different hindsight techniques. HAC, HIRO and HIRO-LR consist of a set of nested policies where the goal of a policy is provided by the top layer. In this setting the goal and state space of the lower policy is identical to the action space of the upper policy. This necessitates sharing of the state space across layers. Overcoming this limitation, we introduce a modular design to decouple the functionality of individual layers. This allows us to define different state, action and goal spaces for each layer. Global information about the environment is only available to the planning layer, while lower levels only receive information that is specific to the respective layer. Furthermore, HAC and HIRO have a state space that includes the agent's position and the goal position, while (Nachum et al., 2019) and our method both have access to global information in the form of a top-down view image.

In model-based reinforcement learning much attention has been given to learning of a dynamics model of the environment and subsequent planning (Sutton, 1990; Sutton et al., 2012; Wang et al., 2019). Eysenbach et al. (2019) propose a planning method that performs a graph search over the replay buffer. However, they require to spawn the agent at different locations in the environment and let it learn a distance function in order to build the search graph. Unlike model-based RL, we do not learn state transitions explicitly. Instead, we learn a spatial value map from collected rewards.

Recently, differentiable planning modules that can be trained via model-free reinforcement learning have been proposed (Tamar et al., 2016; Oh et al., 2017; Nardelli et al., 2019; Srinivas et al., 2018). Tamar et al. (2016) establish a connection between convolutional neural networks and Value Iteration (Bertsekas, 2000). They propose *Value Iteration Networks* (VIN), an approach where model-free RL policies are additionally conditioned on a fully differentiable planning module. MVProp (Nardelli et al., 2019) extends this work by making it more parameter-efficient and generalizable. The planning layer in our approach is based on MVProp, however contrary to prior work we do not rely on a fixed neighborhood mask to sequentially provide actions in its vicinity in order to reach a goal. Instead we propose to learn an attention mask which is used to generate intermediate goals for the underlying layers. Gupta et al. (2017) learn a map of indoor spaces and planning on it using a multi-scale VIN. In their setting, the policy is learned from expert actions using supervised learning. Moreover, the robot operates on discrete set of high level macro actions. Srinivas et al. (2018) propose Universal Planning Networks (UPN), which learn how to plan an optimal action trajectory via a latent space representation. In contrast to our approach, the method relies on expert demonstrations and transfer to harder tasks can only be achieved after retraining.

## 3 BACKGROUND

### 3.1 GOAL-CONDITIONED REINFORCEMENT LEARNING

We model a Markov Decision Process (MDP) augmented with a set of goals $\mathcal{G}$. We define the MDP as a tuple $\mathcal{M} = \{\mathcal{S}, \mathcal{A}, \mathcal{G}, \mathcal{R}, \mathcal{T}, \rho_0, \gamma\}$, where $\mathcal{S}$ and $\mathcal{A}$ are set of states and actions, respectively, $\mathcal{R}_t = r(s_t, a_t, g_t)$ a reward function, $\gamma$ a discount factor $\in [0, 1]$, $\mathcal{T} = p(s_{t+1}|s_t, a_t)$ the transition dynamics of the environment and $\rho_0 = p(s_1)$ the initial state distribution, with $s_t \in \mathcal{S}$ and $a_t \in \mathcal{A}$. Each episode is initialized with a goal $g \in \mathcal{G}$ and an initial state is sampled from $\rho_0$. We aim to find a policy $\pi : \mathcal{S} \times \mathcal{G} \to \mathcal{A}$, which maximizes the expected return.

We train our policies by using an actor-critic framework where the goal augmented action-value function is defined as:

$$Q(s, g, a) = \mathbb{E}_{a_t \sim \pi, s_{t+1} \sim \mathcal{T}} \left[ \sum_{i=t}^{T} \gamma^{i-t} r(s_t, g_t, a_t) | s_t = s, g_t = g, a_t = a \right] \tag{1}$$

The Q-function (critic) and the policy $\pi$ (actor) are approximated by using neural networks with parameters $\theta^Q$ and $\theta^\pi$. The objective for $\theta^Q$ minimizes the loss:

$$L(\theta^Q) = \mathbb{E}_{\mathcal{M}} \left[ \left( Q(s_t, g_t, a_t; \theta^Q) - y_t \right)^2 \right] \quad , \text{where}$$
$$y_t = r(s_t, g_t, a_t) + \gamma Q(s_{t+1}, g_{t+1}, a_{t+1}; \theta^Q) \quad . \tag{2}$$

The policy parameters $\theta^\pi$ are trained to maximize the Q-value:

$$L(\theta^\pi) = \mathbb{E}_\pi \left[ Q(s_t, g_t, a_t; \theta^Q) | s_t, g_t, a_t = \pi(s_t, g_t; \theta^\pi) \right] \tag{3}$$

To address the issue of sparse rewards, we utilize Hindsight Experience Replay (HER) (Andrychowicz et al., 2017), a technique to improve sample-efficiency in training goal-conditioned environments. The insight is that the desired goals of transitions stored in the replay buffer can be relabeled as states that were achieved in hindsight. Such data augmentation allows learning from failed episodes, which may generalize enough to solve the intended goal.

### 3.2 HINDSIGHT TECHNIQUES

In HAC, Levy et al. (2019) apply two hindsight techniques to address the challenges introduced by the non-stationary nature of hierarchical policies and the environments with sparse rewards. In order to train a policy $\pi_i$, optimal behavior of the lower-level policy is simulated by *hindsight action transitions*. More specifically, the action $a_i$ is replaced with a state $s_{i-1}$ that is actually achieved by the lower-level policy $\pi_{i-1}$. Identically to HER, *hindsight goal transitions* replace the subgoal $g_{i-1}$ with an achieved state $s_{i-1}$, which consequently assigns a reward to the lower-level policy $\pi_{i-1}$ for achieving the virtual subgoal. Additionally, a third technique called *subgoal testing* is proposed. The incentive of subgoal testing is to help a higher-level policy understand the current capability of a lower-level policy and to learn Q-values for subgoal actions that are out of reach. We find both techniques effective and apply them to our model during training.

### 3.3 VALUE PROPAGATION NETWORKS

Tamar et al. (2016) propose differentiable value iteration networks (VIN) for path planning and navigation problems. Nardelli et al. (2019) propose value propagation networks (MVProp) with better sample efficiency and generalization behavior. MVProp creates reward- and propagation maps covering the environment. The reward map highlights the goal location and the propagation map determines the propagation factor of values through a particular location. The reward map is an image $\bar{r}_{i,j}$ of the same size as the environment image $I$, where $\bar{r}_{i,j} = 0$ if the pixel $(i, j)$ overlaps with the goal position and $-1$ otherwise. The value map $V$ is calculated by unrolling max-pooling operations in a neighborhood $N$ for $k$ steps as follows:

$$v_{i,j}^{(0)} = \bar{r}_{i,j}$$
$$v_{i,j}^{(k)} = \max \left( v_{i,j}^{(k-1)}, \max_{(i',j') \in N(i,j)} (\bar{r}_{i,j} + p_{i,j}(v_{i',j'}^{(k-1)} - \bar{r}_{i,j})) \right) \tag{4}$$

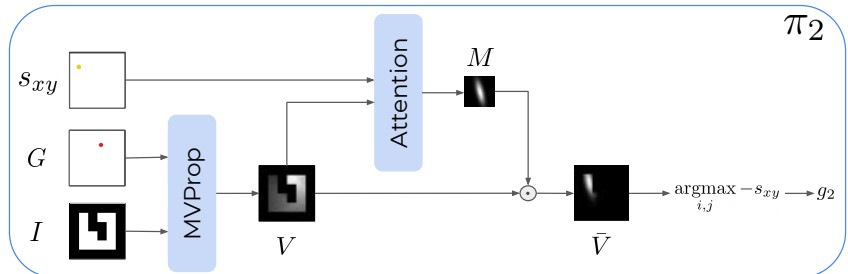

Figure 3: Planner layer $\pi_2(s_{xy}, G, I)$. Given the top-view environment image $I$ and goal $G$ on the map, the maximum value propagation network (MVProp) calculates a value map $V$. By using the agent's current position $s_{xy}$, we estimate an attention mask $M$ restricting the global value map $V$ to a local and reachable subgoal map $\bar{V}$. The policy $\pi_2$ selects the coordinates with maximum value and assigns the lower policy $\pi_1$ with a sugboal that is relative to the agent's current position.

The action (i.e., the target position) is selected to be the pixels $(i', j')$ maximizing the value in a predefined $3x3$ neighborhood $N(i_0, j_0)$ of the agent's current position $(i_0, j_0)$:

$$\pi(s, (i_0, j_0)) = \operatorname*{argmax}_{i', j' \in N(i_0, j_0)} v_{i', j'}^{(k)} \tag{5}$$

Note that the window $N(i_0, j_0)$ is determined by the discrete, pixel-wise actions.

## 4 HIERARCHICAL DECOMPOSITIONAL REINFORCEMENT LEARNING

We introduce a novel hierarchical architecture, HiDe, allowing for an explicit functional *decomposition* across layers. Similar to HAC (Levy et al., 2019), our method achieves temporal abstractions via nested policies. Moreover, our architecture enables functional decomposition explicitly. This is achieved by nesting i) an abstract planning layer, followed ii) by a local planer to iii) guide a control component. Crucially, only the top layer receives global information and is responsible for planning a trajectory towards a goal. The lowest layer learns a control policy for agent locomotion. The middle layer converts the planning layer output into subgoals for the control layer. Achieving functional decoupling across layers crucially depends on reducing the state in each layer to the information that is relevant to its specific task. This design significantly improves generalization (see Section 5).

### 4.1 PLANNING LAYER

The highest layer of a hierarchical architecture is expected to learn high-level actions over a longer horizon, which define a coarse trajectory in navigation-based tasks. In the related work (Levy et al., 2019; Nachum et al., 2018; 2019), the planning layer, learning an *implicit* value function, shares the same architecture as lower layers. Since the task is learned for a specific environment, limits to generalization are inherent to this design choice. In contrast, we introduce a planning specific layer consisting of several components to learn the map and to find a feasible path to the goal.

The planning layer is illustrated in figure 3. We utilize a value propagation network (MVProp) (Nardelli et al., 2019) to learn an *explicit* value map which projects the collected rewards onto the environment image. Given a top-down image of the environment, a convolutional network determines the per pixel flow probability $p_{i,j}$. For example, the probability value of a pixel corresponding to a wall should be $0$ and that for free passages $1$ respectively.

Nardelli et al. (2019) use a predefined $3 \times 3$ neighborhood of the agent's current position and pass the location of the maximum value in this neighbourhood as goal position to the agent (equation 5). We augment a MVProp network with an attention model which learns to define the neighborhood dynamically and adaptively. Given the value map $V$ and the agent's current position $s_{xy}$, we estimate how far the agent can go, modeled by a 2D Gaussian. More specifically, we predict a full covariance matrix $\Sigma$ with the agent's global position $s_{xy}$ as mean. We later build a 2D mask $M$ of the same size as the environment image $I$ by using the likelihood function:

$$m_{i,j} = \mathcal{N}((i, j) | s_{xy}, \Sigma) \tag{6}$$

Figure 4: A visual comparison of *(left)* our dynamic attention window with a *(right)* fixed neighborhood. The green dot corresponds to the selected subgoal in this case. Notice how our window is shaped so that it avoids the wall and induces a further subgoal.

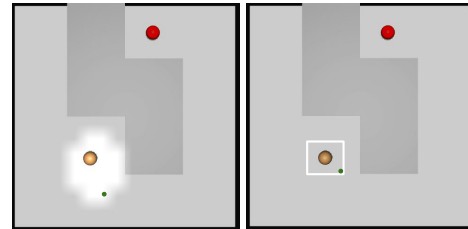

Intuitively, the mask defines the density for the agent's success rate. Our planner policy selects an action (i.e., subgoal) that maximizes the masked value map as follows:

$$\bar{V} = M \cdot V$$
$$\pi_2(s_{xy}, G, I) = \operatorname*{argmax}_{i,j} \bar{v}_{i,j} \qquad (7)$$
$$g_2 = \pi_2(s_{xy}) - s_{xy} \quad ,$$

where $\bar{v}_{i,j}$ corresponds to the value at pixel $(i, j)$ on the masked value map $\bar{V}$. Note that the subgoal selected by the planning layer $g_2$ is relative to the agent's current position $s_{xy}$, which improves generalization performance of our model.

The benefits of having an attention model are twofold. First, the planning layer considers the agent dynamics in assigning subgoals which may lead to fine- or coarse-grained subgoals depending on the underlying agent's performance. Second, the Gaussian window allows us to define a dynamic set of actions for the planner policy $\pi_2$, which is essential to find a trajectory of subgoals on the map. While the action space includes all pixels of the value map $V$, it is limited to the subset of only reachable pixels by the Gaussian mask $M$. Qualitatively we find this leads to better obstacle avoidance behaviour such as the corners and walls shown in figure 4.

Since our planner layer operates in a discrete action space (i.e., pixels), the resolution of the projected maze image defines the minimum amount of displacement for the agent, affecting maneuverability. This could be tackled by using a soft-argmax (Chapelle & Wu, 2010) to select the subgoal pixel, allowing to choose real-valued actions and providing in-variance to image resolution. In our experiments we see no difference in terms of the final performance. However, since the former setting allows for the use of DQN (Mnih et al., 2013) instead of DDPG (Silver et al., 2014), we prefer the discrete action space for simplicity and faster convergence.

Both the MVProp (equation 4) and Gaussian likelihood (equation 6) operations are differentiable. Hence, MVProp and the attention model parameters are trained by minimizing the standard mean squared Bellman error objective as defined in equation 2.

## 4.2 INTERFACE LAYER

The middle layer in our hierarchy interfaces the high-level planning with low-level control by introducing an additional level of temporal abstraction. The planner's longer-term goals are further split into a number of shorter-term targets. Such refinement policy provides the lower-level control layer with reachable targets, which in return yields easier rewards and hence accelerated learning.

The interface layer policy is the only layer that is not directly interacting with the environment. More specifically, the policy $\pi_1$ only receives the subgoal $g_2$ from the upper layer $\pi_2$ and chooses an action (i.e. subgoal $g_1$) for the lower-level locomotion layer $\pi_0$. Note that all the goal, state and action spaces of the policy $\pi_1$ are in 2D space. Contrary to Levy et al. (2019), we use subgoals that are relative to the agent's position $s_{xy}$. This helps to generalize and learn better.

## 4.3 CONTROL LAYER

The lowest layer learns a goal-conditioned control policy. Due to our explicit functional decomposition, it is the only layer with access to the agent's internal state $s_{joints}$ including joint positions and velocities. Whereas the higher layers only have access to the agent's position. In a navigation task, the agent has to learn locomotion to reach the goal position. Similar to HAC, we use *hindsight goal transition* techniques so that the control policy receives rewards even in failure cases.

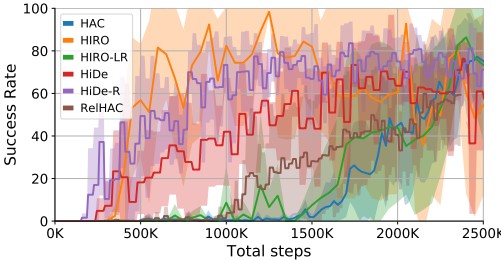

| Experiment 1 | Forward | Backward | Flipped |
|---|---|---|---|
| HAC | $82 \pm 16$ | $0 \pm 0$ | $0 \pm 0$ |
| HIRO | $\mathbf{99 \pm 1}$ | $0 \pm 0$ | $0 \pm 0$ |
| HIRO-LR | $97 \pm 5$ | $0 \pm 0$ | $0 \pm 0$ |
| RelHAC | $66 \pm 27$ | $4 \pm 8$ | $0 \pm 0$ |
| HiDe-R | $89 \pm 3$ | $\mathbf{61 \pm 14}$ | $\mathbf{90 \pm 3}$ |
| HiDe | $85 \pm 6$ | $55 \pm 20$ | $69 \pm 40$ |

Figure 5: Success rates in forward maze wrt the number of environment steps. All algorithms eventually learn the task. HIRO converges the fastest because it benefits from dense rewards. The results are averaged over 5 seeds.

Table 1: Success rates of achieving a goal with an Ant agent in a simple maze environment. All algorithms except for HiDe have been trained with randomly sampled goal positions. The results are averaged over 5 seeds.

All policies in our hierarchy are jointly-trained. We use the DDPG algorithm (Lillicrap et al., 2015) with the goal-augmented actor-critic framework (equation 2-3) for the control and interface layers, and DQN (Mnih et al., 2013) for the planning layer (see section 4.1).

## 5 EXPERIMENTS

We evaluate our method on a series of simulated continuous control tasks in navigation-based environments[1]. All environments are simulated in the MuJoCo physics engine (Todorov et al., 2012). Experiment and implementation details are provided in the Appendix B. First, in section 5.1, we compare to various baseline methods. In section 5.2, we move to a new maze with a more complex design in order to show our model's generalization capabilities. Section 5.3 demonstrates that our approach indeed leads to functional decomposition by composing *new* agents via combining the planning layer of one agent with the locomotion layer of another. Finally, in section 5.4 we provide an ablation study for our design choices. We introduce the following task configurations:

**Maze Forward**: the training environment in all experiments. The task is to reach a goal from a fixed pre-determined start position.
**Maze Backward**: the training maze layout with swapped start and goal positions.
**Maze Flipped**: a mirrored version of the training environment.
**Maze Random**: a set of randomly generated mazes with random start and goal positions.

In our experiments, we always train in the Maze Forward environment. The reward signal during training is constantly -1, unless the agent reaches the given goal (except for HIRO and HIRO-LR, see section 5.1). We test the agents on the above tasks with fixed starting and fixed goal position. For more details about the environments, we refer to Appendix A. We intend to answer the following two questions: 1) Can our method generalize to unseen test environments? 2) Is it possible to transfer the planning layer policies between agents?

### 5.1 SIMPLE MAZE NAVIGATION

We compare our method to state-of-the-art approaches including HIRO (Nachum et al., 2019), HIRO-LR (Nachum et al., 2019), HAC (Levy et al., 2019) and a modified version of HAC called Rel-HAC in a simple Maze Forward environment as shown in figure 6. For a fair comparison, we made a number of improvements to the HAC and HIRO implementations. For HAC, we introduced target networks and used the hindsight experience replay technique with the *future* strategy (Andrychowicz et al., 2017). In our experiments we observed that oscillations around the goal kept HIRO agents from finishing the task, which was solved via doubling the distance-threshold of success. HIRO-LR is the closest to our method, as it also receives a top-down view image of the environment. Note that both HIRO and HIRO-LR have access to dense negative distance reward, which is an advantage over HAC and HiDe that only receive a reward when finishing the task.

---

[1]Videos available at `https://sites.google.com/view/hide-rl`

| Experiment 2 | Ant | Ball |
|---|---|---|
| Forward | $81 \pm 8$ | $96 \pm 7$ |
| Random | $89 \pm 3$ | $96 \pm 1$ |
| Backward | $56 \pm 8$ | $100 \pm 0$ |
| Flipped | $74 \pm 11$ | $99 \pm 2$ |

| Experiment 3 | Ant $\rightarrow$ Ball | Ball $\rightarrow$ Ant |
|---|---|---|
| Forward | $100 \pm 0$ | $66 \pm 14$ |
| Random | $97 \pm 1$ | $86 \pm 5$ |
| Backward | $98 \pm 4$ | $53 \pm 9$ |
| Flipped | $100 \pm 0$ | $59 \pm 27$ |

Table 2: Success rates in the complex maze. We train an ant and a ball on Forward maze and test on unseen Random, Backward, and Flipped environments.

Table 3: Success rates of achieving a goal in complex maze environment with transferred agents. We train with a different agent on Forward maze.

We train a modified HAC model, dubbed RelHAC, to asses our planning layer. RelHAC has the same lowest and middle layers as HiDe, whereas the top layer has the same structure as the middle layer, therefore missing an effective planner. Preliminary experiments using fixed start and fixed goal positions during training for HAC, HIRO and HIRO-LR yielded 0 success rates in all cases. Therefore, the baseline models are trained by using fixed start and random goal positions, allowing it to receive a reward signal without having to reach the intended goal at the other end of the maze. Contrarily, HiDe is trained with fixed start and fixed goal positions, whereas HiDe-R represents HiDe under the same conditions as the baseline methods.

All models learned this task successfully as shown in figure 5 and table 1 (Forward column). HIRO demonstrates slightly better convergence and final performance, which can be attributed to the fact that it is trained with dense rewards. RelHAC performs worse than HAC due to the pruned state space of each layer and due to the lack of an effective planner. HIRO-LR takes longer to converge because it has to learn a latent goal space representation.

Table 1 summarizes the models' generalization abilities to the unseen Maze Backward and Maze Flipped environments (see figure 6). While HIRO, HIRO-LR and HAC manage to solve the training environment (Maze Forward) with success rates between 99% and 82%, they suffer from overfiting to the training environment, indicated by the 0% success rates in the unseen test environments. Contrarily, our method is able to achieve 54% and 69% success rates in this generalization task. As expected, training our model with random goal positions (i.e., HiDe-R) yields a more robust model outperforming vanilla HiDe. In subsequent experiments, we only report the results for our method, as our experiments have shown that the baseline methods cannot solve the training task for more complex environments.

## 5.2 COMPLEX MAZE NAVIGATION

In this experiment, we train an ant and a ball agent (see Appendix A.1) in the Maze Forward task with a more complex environment layout (cf. figure 1), while we keep both the start and goal positions intact. We then evaluate this model in 4 different tasks (see section 5).

Table 2 reports success rates of both agents in this complex task. Our model successfully transfers its navigation skills to unseen environments. The performance for the Maze Backward and Maze Flipped tasks is similar to the results shown in section 5.1 despite the increased difficulty. Since the randomly generated mazes are typically easier, our model shows similar or better performance.

## 5.3 TRANSFER OF POLICIES

To demonstrate that the layers in our architecture indeed learn separate sub-tasks we transfer individual layers across different agents. We first train an agent without our planning layer, i.e., with RelHAC. We then replace the top layer of this agent with the planning layer from the models trained in section 5.2. Additionally, we train a humanoid agent and show as a proof of concept that transfer to a very complex agent can be achieved.

We carry out two sets of experiments. First, we transfer the ant model's planning layer to the simpler 2 DoF ball agent. As indicated in Table 3, the performance of the ball with the ant's planning layer matches the results in Table 2. The ball agent's success rate increases for random (from $96\%$ to $100\%$) and forward ($96\%$ to $97\%$) maze tasks whereas it decreases slightly in the backward (from $100\%$ to $90\%$) and flipped (from $99\%$ to $88\%$) configurations.

| Ant agent | Fwd | Back | Flip |
|---|---|---|---|
| HiDe-A | $0 \pm 0$ | $0 \pm 0$ | $0 \pm 0$ |
| HiDe-AR | $95 \pm 1$ | $52 \pm 33$ | $34 \pm 45$ |

| Ant agent | Fwd | Rand | Back | Flip |
|---|---|---|---|---|
| HiDe-A | $0 \pm 0$ | $0 \pm 0$ | $0 \pm 0$ | $0 \pm 0$ |
| HiDe-AR | $0 \pm 0$ | $0 \pm 0$ | $0 \pm 0$ | $0 \pm 0$ |
| HiDe-NI | $10 \pm 5$ | $46 \pm 16$ | $0 \pm 0$ | $3 \pm 4$ |

Table 4: Success rates in the simple maze. HiDe-A is our method with absolute subgoals. HiDe-AR has absolute goals and samples random goals during training.

Table 5: Success rates of achieving a goal in the complex maze environment. HiDe-A and HiDe-AR as in Table 4. HiDe-NI is our method without the inferface layer.

Second, we attach the ball agent's planning layer to the more complex ant agent. Our new compositional agent performs marginally better or worse in the Flipped, Random and Backward tasks. Please note that this experiment is an example of a case where the environment is first learned with a fast and easy-to-train agent (i.e., ball) and then utilized by a more complex agent. We hereby show that transfer of layers between agents is possible and therefore find our hypothesis to be valid. Moreover, an estimate indicates that the training is roughly *3 – 4 times* faster, since the complex agent does not have to learn the planning layer.

To demonstrate our method's transfer capabilities, we train a humanoid agent (17 DoF) in an empty environment with shaped rewards. We then use the planning and interface layer from a ball agent and connect it as is with the locomotion layer of the trained humanoid[2].

## 5.4 ABLATION STUDIES

To support the claim that our architectural design choices lead to better generalization and functional decomposition, we compare empirical results for different variants of our method with an ant agent. First, we compare the performance of relative and the absolute positions for both experiment 1 and experiment 2. For this reason, we train HiDe-A and HiDe-AR, the corresponding variants of HiDe and HiDe-R that use absolute positions. Unlike for relative positions, the policy needs to learn all values within the range of the environment dimensions. Second, we compare HiDe against a variant of HiDe without the interface layer called HiDe-NI.

The results for experiment 1 are in Table 4. HiDe-A does not manage to solve the task at all, similar to HAC and HIRO without random goal sampling. HiDe-AR succeeds in solving the Forward task. However, it generalizes worse than both Hide and HiDe-R in the Backward and Flipped task. Both HiDe-A and HiDe-AR fail to solve the complex maze for experiment 2 as shown in the Table 5. These results indicate that 1) relative positions improve performance and are an important aspect of our method to achieve generalization to other environments and 2) random goal position sampling can help agents, but may not be available depending on the environment. As seen in Table 5, the variant of HiDe without interface layer (HiDe-NI) performs worse than HiDe (cf. Table 2) in all experiments. Thus, the interface layer is an important part of our architecture.

We also run an ablation study for HiDe with a fixed window size. More specifically, we train and evaluate an ant agent on window sizes $3 \times 3$, $5 \times 5$, and $9 \times 9$. The results are included in Tables 12, 13, and 14. The learned attention window (HiDe) achieves better or comparable performance. In all cases, HiDe generalizes better in the Backward complex maze. Moreover, the learned attention eliminates the need for tuning the window size hyperparameter per agent and environment.

## 6 CONCLUSION

In this paper, we introduce a novel HRL architecture that can solve complex navigation tasks in 3D-based maze environments. The architecture consists of a planning layer which learns an explicit value map and is connected with a subgoal refinement layer and a low-level control layer. The framework can be trained end-to-end. While training with a fixed starting and goal position, our method is able to generalize to previously unseen settings and environments. Furthermore, we demonstrate that transfer of planners between different agents can be achieved, enabling us to transfer a planner trained with a simplistic agent such as a ball to a more complex agent such as an ant or humanoid. In future work, we want to consider integration of a more general planner that is not restricted to navigation-based environments.

---

[2]Videos available at `https://sites.google.com/view/hide-rl`

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

## A  ENVIRONMENT DETAILS

We build on the Mujoco (Todorov et al., 2012) environments used in Nachum et al. (2018). All environments use $dt = 0.02$. Each episode in experiment 1 is terminated after 500 steps and after 800 steps in the rest of the experiments or after the goal in reached. All rewards are sparse as in Levy et al. (2019), i.e., 0 for reaching the goal and $-1$ otherwise. We consider goal reached if $|s - g|_{\max} < 1$. Since HIRO sets the goals in the far distance to encourage the lower layer to move to the goal faster, it can't stay exactly at the target position. Moreover, they do not terminate the episode after the goal is reached. Thus for HIRO, we consider a goal reached if $|s - g|_2 < 2.5$.

### A.1  AGENTS

Our Ant agent is equivalent to the one in Levy et al. (2019). In other words, the Ant from Rllab (Duan et al., 2016) with gear power of 16 instead of 150 and 10 frame skip instead of 5. Our Ball agent is the PointMass agent from DM Control Suite (Tassa et al., 2018). We made the change the joints so that the ball rolls instead of sliding. Furthermore, we resize the motor gear and the ball itself to match the maze size.

### A.2  MAZES

All mazes are modelled by immovable blocks of size $4 \times 4 \times 4$. Nachum et al. (2018) uses blocks of $8 \times 8 \times 8$. The environment shapes are clearly depicted in figure 1. For the randomly generated maze, we sample each block with probability being empty $p = 0.8$, start and goal positions are also sampled randomly at uniform. Mazes where start and goal positions are adjacent or where goal is not reachable are discarded. For the evaluation, we generated 500 of such environments and reused them (one per episode) for all experiments.

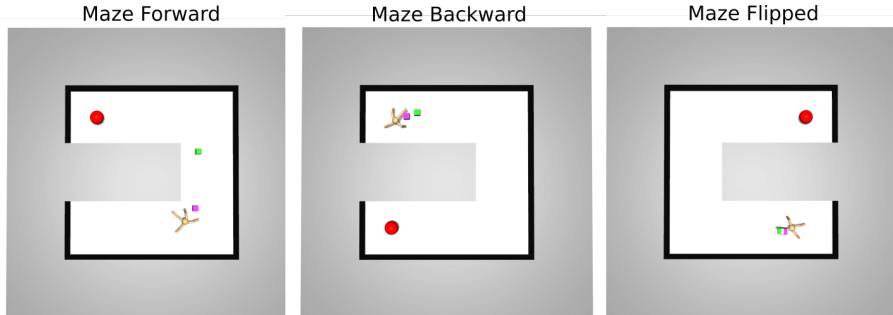

Figure 6: Environments used for testing in Section 5.1. The red sphere indicates the goal of the task.

## B  IMPLEMENTATION DETAILS

Our PyTorch (Paszke et al., 2017) implementation will be available at the project website. [3]

### B.1  BASELINE EXPERIMENTS

For both HIRO and HAC we used the original authors implementation[4][5]. In HIRO, we set the goal success radius for evaluation as described above. We ran the hiro_xy variant, which uses only position coordinates for subgoal instead of all joint positions, to have a fair comparison with our method. To improve the performance of HAC in experiment one, we modified their Hindsight Experience Replay (Andrychowicz et al., 2017) implementation so that they use FUTURE strategy. More importantly, we also added target networks to both the actor and critic to improve the performance.

---

[3] https://sites.google.com/view/hide-rl

[4] HIRO: https://github.com/tensorflow/models/tree/master/research/efficient-hrl

[5] HAC: https://github.com/andrew-j-levy/Hierarchical-Actor-Critc-HAC-

## B.2 EVALUATION DETAILS

For evaluation, we trained 5 seeds each for 2.5M steps on the Forward environment with continuous evaluation (every 100 episodes for 100 episodes). After training, we selected the best checkpoint based on the continuous evaluation of each seed. Then, we tested the learned policies for 500 episodes and reported the average success rate. Although the agent and goal positions are fixed, the initial joint positions and velocities are sampled from uniform distribution as standard in OpenAI Gym environments. Therefore, the tables in the paper contain means and standard deviation across 5 seeds.

## B.3 NETWORK STRUCTURE

### B.3.1 PLANNING LAYER

Input images for the planning layer were binnarized in the following way: each pixel corresponds to one block (0 if it was a wall or 1 if it was a corridor). In our planning layer, we process the input image of size 32x32 (20x20 for experiment 1) via two convolutional layers with $3 \times 3$ kernels. Both layers have only 1 input and output channel and are padded so that the output size is the same as the input size. We propagate the value through the value map as in Nardelli et al. (2019) $K = 35$ times using a $3 \times 3$ max pooling layer. Finally, the value map and agent position image (a black image with a dot at agent position) is processed by 3 convolutions with 32 output channels and $3 \times 3$ filter window interleaved by $2 \times 2$ max pool with $\mathrm{ReLU}$ activation functions and zero padding. The final result is flatten and processed by two fully connected layers with $64$ neurons each producing three outputs: $\sigma_1, \sigma_2, \rho$ with $\mathrm{softplus}$, $\mathrm{softplus}$ and $\mathrm{tanh}$ activation functions respectively. The final covariance matrix $\Sigma$ is given by

$$\Sigma = \begin{pmatrix} \sigma_1^2 & \rho\,\sigma_1\sigma_2 \\ \rho\,\sigma_1\sigma_2 & \sigma_2^2, \end{pmatrix}$$

so that the matrix is always symmetric and positive definite. For numerical reasons, we multiply by binnarized kernel mask instead of the actual Gaussian densities. We set values higher than mean to 1 and others to zeros. In practice, we use this line:

```
kernel = t.where(kernel >= kernel.mean(dim=[1,2], keepdim=True),
                 t.ones_like(kernel), t.zeros_like(kernel))
```

### B.3.2 MIDDLE AND LOCOMOTION LAYER

We use the same network architecture for the middle and lower layer as proposed by Levy et al. (2019), i.e. we use 3 times fully connected layer with $\mathrm{ReLU}$ activation function. The locomation layer is activated with $\mathrm{tanh}$, which is then scaled to the action range.

### B.3.3 TRAINING PARAMETERS

- Discount $\gamma = 0.98$ for all agents.
- Adam optimizer. Learning rate $0.001$ for all actors and critics.
- Soft updates using moving average; $\tau = 0.05$ for all controllers.
- Replay buffer size was designed to store 500 episodes, similarly as in Levy et al. (2019)
- We performed $40$ actor and critic learning updates after each epoch on each layer, after the replay buffer contained at least 256 transitions.
- Batch size 1024.
- No gradient clipping
- Rewards 0 and -1 without any normalization.
- Subgoal testing (Levy et al., 2019) only for the middle layer.
- Maximum subgoal horizon $H = 10$ for all 3 layers algorithms and $H = 25$ for ablations without the inferace layer. See psuedocode 1.
- Observations also were not normalized.

- 2 HER transitions per transition using the FUTURE strategy (Andrychowicz et al., 2017).
- Exploration noise: 0.05, 0.01 and 0.1 for the planning, middle and locomotion layer respectively.

## C ADDITIONAL RESULTS

In this section, we present all results collected for this paper including individual runs.

| Experiment | Forward | Backward | Flipped |
|---|---|---|---|
| HAC 1 | 96.4 | 00.0 | 00.0 |
| HAC 2 | 82.0 | 00.0 | 00.0 |
| HAC 3 | 85.6 | 00.4 | 00.0 |
| HAC 4 | 92.8 | 00.0 | 00.0 |
| HAC 5 | 55.6 | 00.0 | 00.0 |
| HIRO 1 | 100 | 00.0 | 00.0 |
| HIRO 2 | 99.8 | 00.0 | 00.0 |
| HIRO 3 | 99.0 | 00.0 | 00.0 |
| HIRO 4 | 99.6 | 00.0 | 00.0 |
| HIRO 5 | 97.2 | 00.0 | 00.0 |
| HIRO-LR 1 | 88 | 00.0 | 00.0 |
| HIRO-LR 2 | 100 | 00.0 | 00.0 |
| HIRO-LR 3 | 97.6 | 00.0 | 00.0 |
| HIRO-LR 4 | 98.8 | 00.0 | 00.0 |
| HIRO-LR 5 | 100 | 00.0 | 00.0 |
| RelHAC 1 | 77.6 | 18.8 | 00.0 |
| RelHAC 2 | 86.4 | 00.0 | 00.0 |
| RelHAC 3 | 77.8 | 00.0 | 00.0 |
| RelHAC 4 | 19.6 | 00.0 | 00.0 |
| RelHAC 5 | 71.0 | 00.0 | 00.0 |
| HiDe-R 1 | 87.6 | 72.2 | 90.6 |
| HiDe-R 2 | 87.8 | 70.4 | 89.6 |
| HiDe-R 3 | 86.4 | 38.2 | 89.8 |
| HiDe-R 4 | 90.6 | 69.4 | 94.0 |
| HiDe-R 5 | 94.4 | 56.2 | 87.0 |
| HiDe 1 | 80.6 | 21.2 | 00.0 |
| HiDe 2 | 94.6 | 71.8 | 96.8 |
| HiDe 3 | 81.6 | 53.4 | 90.2 |
| HiDe 4 | 79.4 | 61.4 | 91.2 |
| HiDe 5 | 87.0 | 66.0 | 66.6 |

Table 6: Results for experiment 1 for individual seeds.

| | Ant 1 | Ant 2 | Ant 3 | Ant 4 | Ant 5 | Ball 1 | Ball 2 | Ball 3 | Ball 4 | Ball 5 |
|---|---|---|---|---|---|---|---|---|---|---|
| Forward | 82.0 | 80 | 92.2 | 70.6 | 82.0 | 99.6 | 99.0 | 100 | 83.2 | 100 |
| Random | 85.8 | 88.2 | 89.2 | 91.4 | 92.0 | 97.0 | 94.2 | 97.6 | 97.2 | 96.0 |
| Backward | 61.8 | 61.4 | 56.0 | 42.0 | 60.4 | 100 | 100 | 100 | 100 | 100 |
| Flipped | 87.4 | 68.2 | 64.8 | 64.0 | 85.4 | 100 | 96.4 | 100 | 100 | 100 |

Table 7: Results for experiment 2 for individual seeds.

|          | A→B 1 | A→B 2 | A→B 3 | A→B 4 | A→B 5 |
|----------|-------|-------|-------|-------|-------|
| Forward  | 100   | 100   | 100   | 100   | 100   |
| Random   | 96.2  | 96.6  | 96.8  | 97.8  | 96.6  |
| Backward | 90.2  | 99.8  | 99.8  | 99.2  | 100   |
| Flipped  | 100   | 100   | 100   | 100   | 100   |

Table 8: Results for Ant to Ball transfer for individual seeds.

|          | B→A 1 | B→A 2 | B→A 3 | B→A 4 | B→A 5 |
|----------|-------|-------|-------|-------|-------|
| Forward  | 45.8  | 81.2  | 75.8  | 64.8  | 64.4  |
| Random   | 81.6  | 85    | 94.4  | 83.4  | 88.0  |
| Backward | 49.4  | 45.4  | 68.6  | 48.0  | 54.6  |
| Flipped  | 32.0  | 83.8  | 79.6  | 26.8  | 70.4  |

Table 9: Results for Ball to Ant transfer for individual seeds.

|          | Ant 1 | Ant 2 | Ant 3 | Ant 4 | Ant 5 | Averaged    |
|----------|-------|-------|-------|-------|-------|-------------|
| Forward  | 94.0  | 96.0  | 96.2  | 95.0  | 93.4  | $95 \pm 1$  |
| Backward | 84.2  | 40.6  | 1.2   | 59.4  | 76.4  | $52 \pm 33$ |
| Flipped  | 2.0   | 1.8   | 75.4  | 90.4  | 0.0   | $34 \pm 45$ |

Table 10: Results for experiment 1 on HiDe-AR.

|          | Ant 1 | Ant 2 | Ant 3 | Ant 4 | Ant 5 | Averaged    |
|----------|-------|-------|-------|-------|-------|-------------|
| Forward  | 7.0   | 12.6  | 6.2   | 16.8  | 6.2   | $10 \pm 5$  |
| Random   | 29.0  | 40.2  | 37.6  | 67.2  | 57.8  | $46 \pm 16$ |
| Backward | 0.0   | 0.2   | 0.0   | 0.8   | 0.0   | $0 \pm 0$   |
| Flipped  | 0.0   | 3.6   | 0.0   | 9.4   | 0.0   | $3 \pm 4$   |

Table 11: Results for experiment 2 on HiDe without interface layer.

|          | Ant 1 | Ant 2 | Ant 3 | Ant 4 | Ant 5 | Averaged    |
|----------|-------|-------|-------|-------|-------|-------------|
| Forward  | 38.6  | 49.8  | 42.2  | 75.6  | 33.8  | $48 \pm 17$ |
| Random   | 69.4  | 83.8  | 70.8  | 86.4  | 64.2  | $75 \pm 10$ |
| Backward | 7.2   | 55.4  | 25.4  | 72.6  | 0.0   | $32 \pm 31$ |
| Flipped  | 0.0   | 69.8  | 0.0   | 0.0   | 0.0   | $14 \pm 31$ |

Table 12: Results for experiment 2 with fixed 3x3 attention window.

|          | Ant 1 | Ant 2 | Ant 3 | Ant 4 | Ant 5 | Averaged    |
|----------|-------|-------|-------|-------|-------|-------------|
| Forward  | 89.0  | 88.0  | 78.8  | 96.4  | 86.6  | $88 \pm 6$  |
| Random   | 87.8  | 93.0  | 89.2  | 92.0  | 87.0  | $90 \pm 3$  |
| Backward | 58.2  | 73.6  | 45.2  | 0.0   | 3.2   | $36 \pm 33$ |
| Flipped  | 59.0  | 84.0  | 46.4  | 0.0   | 81.0  | $54 \pm 34$ |

Table 13: Results for experiment 2 with fixed 5x5 attention window.

|          | Ant 1 | Ant 2 | Ant 3 | Ant 4 | Ant 5 | Averaged    |
|----------|-------|-------|-------|-------|-------|-------------|
| Forward  | 92.0  | 75.4  | 80.2  | 91.0  | 94.6  | $87 \pm 8$  |
| Random   | 84.2  | 83.4  | 85.0  | 91.2  | 89.2  | $87 \pm 3$  |
| Backward | 6.4   | 48.2  | 55.2  | 85.0  | 29.8  | $45 \pm 29$ |
| Flipped  | 85.2  | 64.2  | 81.6  | 93.6  | 71.8  | $79 \pm 12$ |

Table 14: Results for experiment 2 with fixed 9x9 attention window.

---

**Algorithm 1** Hierarchical Decompositional Reinforcement Learning (HiDe)

---

**Input:**

- Agent position $s_{xy}$, goal position $g_{xy}$, and projection from environment coordinates to image coordinates and its inverse $Proj, Proj^{-1}$.

**Parameters:**

1. maximum subgoal horizon $H = 10$, subgoal testing frequency $\lambda = 0.3$

**Output:**

- $k = 3$ trained actor and critic functions $\pi_0, ..., \pi_{k-1}, Q_0, ..., Q_{k-1}$

**for** $M$ episodes **do**                                                      ▷ Train for M episodes
    $s \leftarrow S_{init}, g \leftarrow G_{k-1}$                       ▷ Get initial state and task goal
    $train\_top\_level(s, g)$                                          ▷ Begin training
    Update all actor and critic networks
**end for**

**function** $\pi_2(s :: state, g :: goal)$
    $v_{map} \leftarrow MVProp(I, g_2)$          ▷ Run MVProp on top-down view image and goal position
    $\sigma_1, \sigma_2, \rho \leftarrow CNN(v_{map}, Proj(s_{xy}))$          ▷ Predict mask parameters
    $\Sigma = [\sigma_1^2, \quad \sigma_1\sigma_2\rho, \quad \sigma_1\sigma_2\rho, \quad \sigma_2^2]$
    $v \leftarrow v_{map} \odot \mathcal{N}(\cdot | s_{xy}, \Sigma)$          ▷ Mask the value map
    **return** $a_2 \leftarrow Proj^{-1}(\arg\max v) - s_{xy}$ ▷ Output relative subgoal corresponding to the max value pixel
**end function**

**function** $\pi_1(s :: state, g :: relative\_subgoal)$
    **return** $a_1 \leftarrow MLP(g)$                                   ▷ Output fine-grained relative subgoal
**end function**

**function** $\pi_0(s :: joints\_state, g :: relative\_subgoal)$
    **return** $a_0 \leftarrow MLP(s, g)$                               ▷ Output actions for actuators
**end function**

**function** TRAIN_LEVEL($i :: level, s :: state, g :: goal$)
    $s_i \leftarrow s, g_i \leftarrow g$                                 ▷ Set current state and goal for level $i$
    **for** $H$ attempts or until $g_n, i \leq n < k$ achieved **do**
        $a_i \leftarrow \pi_i(s_i, g_i) + noise$ (if not subgoal testing)          ▷ Sample (noisy) action from policy
        **if** $i > 0$ **then**
            Determine whether to test subgoal $a_i$
            $s_i^{'} \leftarrow train\_level(i - 1, s_i, a_i)$                 ▷ Train level $i - 1$ using subgoal $a_i$
        **else**
            Execute primitive action $a_0$ and observe next state $s_0^{'}$
        **end if**
                                                            ▷ Create replay transitions
        **if** $i > 0$ and $a_i$ not reached **then**
            **if** $a_i$ was subgoal tested **then**                          ▷ Penalize subgoal $a_i$
                $Replay\_Buffer_i \leftarrow [s = s_i, a = a_i, r = Penalty, s^{'} = s_i^{'}, g = g_i, \gamma = 0]$
            **end if**
            $a_i \leftarrow s_i^{'}$                          ▷ Replace original action with action executed in hindsight
        **end if**
                                            ▷ Evaluate executed action on current goal and hindsight goals
        $Replay\_Buffer_i \leftarrow [s = s_i, a = a_i, r \in \{-1, 0\}, s^{'} = s_i^{'}, g = g_i, \gamma \in \{\gamma, 0\}]$
        $HER\_Storage_i \leftarrow [s = s_i, a = a_i, r = TBD, s^{'} = s_i^{'}, g = TBD, \gamma = TBD]$
        $s_i \leftarrow s_i^{'}$
    **end for**
    $Replay\_Buffer_i \leftarrow$ Perform HER using $HER\_Storage_i$ transitions
    **return** $s_i^{'}$                                                 ▷ Output current state
**end function**

---

