# OpenReview forum: "Learning Functionally Decomposed Hierarchies for Continuous Navigation Tasks"
_ICLR.cc/2020/Conference — Reject_

### Official Review · AnonReviewer2 · 2019-10-17
**Official Blind Review #2**

**Rating:** 6

**Review:**

The paper proposes a neat framework for creating HRL framework that will be able to generalize its application to slightly different environment layout. This is done via an image-based top-down from as input to the high level. An intermediate layer is used to help create more fine-grained goal specification for a final goal-based control layer. These layers are trained together using HAC. Overall, the method shows promise but there needs to be more analysis to understand which parts of this combination of ideas are the most important. The results are also only shown for a single environment. Last, the generalization analysis in the paper does not appear to be overly thorough. It would be good to perform this on more than one type of environment also the random environment is not very random.

More detailed comments:
- The results seem very similar to some of the work in "Universal Planning Networks" that did not need a more complex HRL design to achieve subgoal specification via images. This should be discussed more in the paper.
- The authors point out that the use of relative goal positions "ensures generalization to new environments", this is a rather strong statement. The use of relative goal specification my help improve generalization but that can only be shown empirically.
- The demonstration to show that the method generalizes to other configuration after being trained on a fixed environment should be evaluated over many randomly generated environments so that we have a non-biased estimate of the true generalization performance. In Table 2, it is rather surprising that the HiDe trained model does better on the "Random" environment vs the "FOrward" environment it is trained on. Can more details be provided on how the "Random" environment is created? Are the locations of the walls randomized? Is the initial position of the agent and goal randomize?
- The video seems to contradict the ordering of operations for training the planning network. The video suggests that first it is learned with the Ant then transferred to the ball which is less complex to control.
- At the beginning of the prior work section, it is noted that many other methods require prior knowledge of the environment I would say this method also requires certain kinds of prior knowledge about the task. For example, a top-down view of the environment is needed which is not often feasible.
- HIRO and HAC use a more proprioceptive state space but I don't think the sharing of global states is intentional. I am not convinced that this choice, in particular, is what makes the approaches prone to overfitting.
- You show a comparison to the "windows" created from your method vs a fixed neighbourhood. Do you perform any empirical evidence that your introduced methods provide an improvement over this fixed window?
- It is mentioned at the end of section 4.1 that MVProp is differential so it can be trained with the Bellman error objective. Because many policies are being trained concurrently does the MVProp attention model need to be recomputed after every sub-policy update? Does the frequency of updates have a large effect on performance?
- Section 4.2 introduces an interface layer that is not a very common practice. It would be good to include an ablation study of the effects of this introduced layer.
- In section 4.3 it says that the control layer is the only layer with access to the agent's proprioceptive state. Would it not be good to at least include the agent facing direction or current average velocity to higher layers to improve the attention mask estimation?
- In figure 5 it says HIRO converges the fastest because it has dense rewards. Can you be more specific? Also, If different agents are using different reward signals I am not sure this evaluation is a fair comparison.
- Are tables 2 and 3 just for the HiDe algorithm? Is it possible to include data for the other algorithms?
- You perform an experiment to train HiDe with random initial and goal locations for comparison. I think running this comparison for HIRO and HAC would be a good additional point of comparison. This would help the reader know if the generalization is not biased to the particular initial environment configuration for Maze Forward.
- In the generalization analysis for the paper, how is the analysis performed? There are percentages for the success of the policy, where does the randomness come from is the agent state and goal are always fixed? Are these averaged because the agent has a stochastic policy during evaluation? If this is the case how many random trajectories are collected to compute these statistics?
- It would be very helpful to have a description of the algorithm in the paper. How the algorithm works is not very clear and some details about how the goal and states are passed to the different policies would be very helpful if anyone wanted to reimplement this work.

Updated comments:

- The addition of more results and added analysis helps show the improvement of this method over the most related baselines.

**Experience Assessment:**

I have published in this field for several years.

**Review Assessment: Checking Correctness Of Derivations And Theory:**

I carefully checked the derivations and theory.

**Review Assessment: Checking Correctness Of Experiments:**

I carefully checked the experiments.

**Review Assessment: Thoroughness In Paper Reading:**

I read the paper thoroughly.

---

> ### Author Response · Authors · 2019-11-15
> **Improvements, Clarifications and Ablations**
>
> We thank the reviewer for the extensive feedback. We hope we can address most of the reviewer’s concerns.
>
> *** Comparison to Universal Planning Networks ***
> Comment: The results seem very similar to some of the work in "Universal Planning Networks" that did not need a more complex HRL design to achieve subgoal specification via images. This should be discussed more in the paper.
>
> We have added a more detailed comparison between our work and Universal Planning Networks [1] to the updated version of the paper. While we agree that UPN shows similar results in navigation domains, we argue that the problem  we are solving is different. Our agents can learn solely from a sparse binary reward, which means it only receives 0 if it reaches the final goal and -1 in all other cases. Hence, the agent does not rely on a shaped reward function to facilitate learning or demonstrations as a guidance signal. Contrarily, UPN optimizes a supervised imitation objective, i.e., uses demonstrations to help agents learn a task. Acquiring the necessary ground-truth data may not always be feasible.
>
> *** Relative Goal Positions ***
> Comment: The authors point out that the use of relative goal positions "ensures generalization to new environments", this is a rather strong statement. The use of relative goal specification my help improve generalization but that can only be shown empirically.
>
> We rephrase the rather strong statement of “ensures generalization” to “encourages generalization”.  To empirically show this, we provide an ablation study comparing absolute against relative positions (see general comment for the results and discussion).
>
> *** Comments about the Random Environment***
> 1. Comment: Can more details be provided on how the "Random" environment is created? Are the locations of the walls randomized? Is the initial position of the agent and goal randomize?
>
> 2. Comment: The demonstration to show that the method generalizes to other configuration after being trained on a fixed environment should be evaluated over many randomly generated environments so that we have a non-biased estimate of the true generalization performance.
>
> We refer the reviewer to appendix A.2 of the initial submission, where the procedure of creating the environments is described in detail. To summarize, we create 500 randomly generated environments with randomly sampled wall locations, start and goal positions. We then test on the same 500 random environments for all seeds to ensure a fair comparison, so it is an unbiased estimate of the true generalization performance (addressing comment 2.).
>
> 3. Comment: In Table 2, it is rather surprising that the HiDe trained model does better on the "Random" environment vs the "Forward" environment it is trained on.
>
> The reason why the results are (sometimes) better compared to the forward task is because the start and goal positions can essentially be closer together due to the random sampling, while in the forward task the agent always has to move around the whole map to reach the goal.
>
> *** Transfer Experiment Videos ***
> Comment: The video seems to contradict the ordering of operations for training the planning network. The video suggests that first it is learned with the Ant then transferred to the ball which is less complex to control.
>
> In the video as well as in the results of the initial submission (cf. Table 3), we provide both the transfer of the planning layer from the ball to the more complex ant agent ( https://drive.google.com/file/d/1mJhZPmGsr1n-JjSNBn42ExUzVjz41Nek/view?t=1m45s ) and vice-versa ( https://drive.google.com/file/d/1mJhZPmGsr1n-JjSNBn42ExUzVjz41Nek/view?t=1m27s ).
>
> *** Prior Knowledge ***
> Comment: It is noted that many other methods require prior knowledge of the environment I would say this method also requires certain kinds of prior knowledge about the task. For example, a top-down view of the environment is needed which is not often feasible.
>
> We agree on this and therefore have removed the part-sentence about prior knowledge from the related work section. Moreover, we now provide results that compare our approach against [3], which is follow-up work of HIRO that also uses the top-down view image of the environment (see general comment for the results and discussion).

---

> > ### Author Response · Authors · 2019-11-15
> > **Improvements, Clarifications and Ablations Part 2**
> >
> > *** Comparison to HAC and HIRO***
> > 1. Comment: You perform an experiment to train HiDe with random initial and goal locations for comparison. I think running this comparison for HIRO and HAC would be a good additional point of comparison. This would help the reader know if the generalization is not biased to the particular initial environment configuration for Maze Forward.
> >
> > We first want to clarify possible misunderstandings for experiment 1. We always train HAC and HIRO with a fixed starting and random goal position. Testing is performed with a fixed start and fixed goal position. When we trained with a fixed goal position, HAC and HIRO never managed to solve even the “Forward” task in experiment 1. We then compare against our approach trained in the same setting (HiDe-R). Moreover, we train our method in the more constrained setting of a fixed goal position (HiDe) that lies at the other end of the environment. The algorithm always receives -1 as reward until it reaches the final goal. Hence, there needs to be sufficient exploration to reach the final goal at all, which we achieve through our architecture (contribution 1). We show that even within this constrained training configuration, our method can generalize to unseen environments and random initial configurations (contribution 2). Thus, it does not overfit to the fixed starting and goal position.
> >
> > 2. Comment: Are tables 2 and 3 just for the HiDe algorithm? Is it possible to include data for the other algorithms?
> >
> > For experiment 2, we ran experiments for HAC and HIRO, but they never managed to solve the task, even for the “Forward” environment. Therefore, we chose not to include the results in the paper. However, we have added a sentence stating this in the updated paper. An example of HIRO failing can be seen in the video at 0:55 ( https://drive.google.com/file/d/1mJhZPmGsr1n-JjSNBn42ExUzVjz41Nek/view?t=0m55s ). Since HIRO uses negative distance rewards, it never learns to find a way around the walls and just tries to push through to reach the goal. HAC also fails to solve the task, because it never receives successful reward as its exploration capabilities are not sufficient. The results for the ant in experiment 2 look as follows:
> >
> > ======================================================
> > 			Forward		Random		Backward	Flipped
> > HAC		0+-0		0+-0		0+-0		0+-0
> > HIRO		0+-0		0+-0		0+-0		0+-0
> > HiDe		81+-8		89+-3		56+-8		74+-11
> >
> > ======================================================
> > If wished by the reviewer, we will gladly add these scores to the paper.
> >
> > For experiment 3, it is not possible to use neither HAC nor HIRO for transfer, as they share the same state space in all layers, not allowing to decouple the layers out of their trained structure.
> >
> > 3. Comment: HIRO and HAC use a more proprioceptive state space but I don't think the sharing of global states is intentional. I am not convinced that this [...] is what makes the approaches prone to overfitting.
> >
> > We politely disagree and believe that it is intentional, as it is mentioned in [2] that “The state space for every level i is identical to the state space in the original problem: Si=S” and also described in [3] Section 3.1. Proving that this is what makes the approaches prone to overfitting is difficult. However, our results show that removing this shared structure along with other changes leads to better performance. Moreover, it allows transfer of policies between different agents.
> >
> > 4. Comment: In figure 5 it says HIRO converges the fastest because it has dense rewards. Can you be more specific? Also, If different agents are using different reward signals I am not sure this evaluation is a fair comparison.
> >
> > HIRO has access to a dense reward signal, i.e., it can benefit from a reward that consists of the negative distance between the agent and the goal at each timestep and is therefore more privileged. HAC and HiDe, on the other hand, only receive a reward once the actual goal is reached, which makes the task harder.
> >
> > *** Interface Layer Ablation ***
> > Comment: Section 4.2 introduces an interface layer that is not a very common practice. It would be good to include an ablation study of the effects of this introduced layer.
> >
> > The interface layer is motivated by results from previous work [2]. Furthermore, it is also claimed by [3] that the use of more layers leads to better exploration. For results, see general comment.
> >
> > *** Algorithm Description and Code ***
> > Comment: It would be very helpful to have a description of the algorithm in the paper. [...] it would be very helpful if anyone wanted to reimplement this work.
> >
> >  We have added a description of the algorithm to the revised version of the paper. Additionally, all the code as well as pretrained models have been publicly released with the initial submission under an open licence.

---

> > > ### Author Response · Authors · 2019-11-15
> > > **Improvements, Clarifications and Ablations Part 3**
> > >
> > > *** Planning Layer ***
> > > 1. Comment: Does the MVProp attention model need to be recomputed after every sub-policy update? Does the frequency of updates have a large effect on performance?
> > >
> > > No, it does not need to be recomputed after every sub-policy update. Only if the sub-policy achieves the goal or if the subpolicy runs out of attempts (10 in our case). Since a new convolutional network needs to be trained to analyze the top-down view, more compute is required.
> > >
> > > 2. Comment: Do you perform any empirical evidence that your introduced methods provide an improvement over this fixed window?
> > >
> > > To address this, we conducted an additional experiment comparing fixed window sizes to our learned window size for the ant agent. A fixed window of 3x3 as proposed in [4] performs worse than our learned attention window. The size of 5x5 window shows similar performance on the “Forward” and “Random” task, but does not generalize as well to “Backward” and “Flipped”. The size of 9x9 is competitive with ours. The advantage of a learned window is that the window size parameter does not need to be tuned to the agent or environment. The results are as follows:
> > >
> > > 							Forward		Random		Backward	Flipped
> > > Fixed Window 3x3			48+-16		75+-10		32+-31		14+-31
> > > Fixed Window 5x5			88+-6		90+-3		36+-33		54+-34
> > > Fixed Window 9x9			87+-8		87+-3		45+-29		79+-11
> > > Ours					        81+-8		90+-2		56+-10		74+-11
> > >
> > > *** Generalization Analysis ***
> > > Comment:  In the generalization analysis for the paper, how is the analysis performed? There are percentages for the success of the policy, where does the randomness come from is the agent state and goal are always fixed? Are these averaged because the agent has a stochastic policy during evaluation? If this is the case, how many random trajectories are collected to compute these statistics?
> > >
> > > For evaluation, we trained 5 seeds each for 2.5M steps on the “Forward” environment with continuous evaluation (every 100 episodes for 100 episodes). After training, we selected the best checkpoint based on the continuous evaluation of each seed. Then, we tested the learned policies for 500 episodes and reported the average success rate. Although the agent and goal positions are fixed, the initial joint positions and velocities are sampled from uniform distribution as standard in OpenAI Gym environments. Therefore, the tables in the paper contain means and standard deviation across 5 seeds. We have added this information to the revised version of the paper.
> > >
> > > *** Possible Improvement in State Space Choice ***
> > >
> > > Comment: In section 4.3 it says that the control layer is the only layer with access to the agent's proprioceptive state. Would it not be good to at least include the agent facing direction or current average velocity to higher layers to improve the attention mask estimation?
> > >
> > > This is a good point. Such an addition to the planning layer might improve the attention mask. We consider adding this to our approach in the future.
> > >
> > > *** References ***
> > > [1] Aravind Srinivas, Allan Jabri, Pieter Abbeel, Sergey Levine, and Chelsea Finn.  Universal planning networks, abs/1804.00645, 2018.
> > > [2] Andrew Levy, Robert Platt, and Kate Saenko. Learning Multi-Level Hierarchies with Hindsight. InInternational Conference on Learning Representations, 2019.
> > > [3] Anonymous Authors, Why Does Hierarchy (Sometimes) Work So Well in Reinforcement Learning?, under double blind review for ICLR 2020, https://openreview.net/forum?id=rJgSk04tDH
> > > [4] Nantas Nardelli, Gabriel Synnaeve, Zeming Lin, Pushmeet Kohli, Philip H. S. Torr, and NicolasUsunier. Value Propagation Networks. InInternational Conference on Learning Representations,2019

---

### Official Review · AnonReviewer1 · 2019-10-21
**Official Blind Review #1**

**Rating:** 6

**Review:**

This paper addresses hierarchical deep reinforcement learning (RL), an important problem in control learning and RL. Based on my understanding of this paper and recent prior work, the most important difference between the proposed approach (HiDe) and other recent approaches, such as HIRO and HAC, is that the top-level goal proposal policy uses a learned planner based on VIN and a learned attention mask to decide on a subgoal. There are also other differences, e.g., this policy outputs a goal position that is relative to the agent's position, rather than an absolute position. HiDe seems to demonstrate impressive transferability to both unseen mazes and new agent embodiments, which are important problems to address for hierarchical RL.

Introducing learned planning and attention into the top-level policy seems to also introduce additional assumptions into the method. For example, it is my understanding that HIRO and HAC use the same state representation, e.g., joint positions and velocities of the agent, as input to their top-level policies. In contrast, HiDe uses a top down view of the maze and the x y position of the agent, which certainly is more privileged information. If this is correct, first, this should be discussed more thoroughly and directly in the paper. Second, the experimental setup should be elaborated on: is HIRO or HAC modified to include the same information for the top-level policy? Or can HiDe somehow be extended to not require this information? I do not think that requiring this information is egregious, but currently the experimental comparison is not clear in this regard.

The experiments are arguably the strongest part of the paper, and the transfer results and videos are quite nice. But there is still room for improvement. Table 1 and Figure 5 seem disconnected. In particular, the numbers reported in Table 1 are clearly not achieved in Figure 5. Is the figure cut off early? Furthermore, an additional experiment on a more complicated domain would greatly strengthen the paper. A humanoid agent, for example, seems easy to test for the current method. Another option would be the movable blocks tested in HIRO, though it is unclear if this readily fits into the current method's assumptions. In my opinion, as the paper currently rests heavily on the results, this section should be further improved. Doing so would also improve my rating of the paper.

------
Edit after author response: I appreciate the authors' efforts in providing extensive responses to all of the reviewers' concerns as well as a significant general response detailing what seems to be a large amount of additional experimental work. I think that all of this warrants a change to my score, and seeing that the authors have more or less addressed my concerns, I am bumping up to a weak accept.

**Experience Assessment:**

I have read many papers in this area.

**Review Assessment: Checking Correctness Of Derivations And Theory:**

N/A

**Review Assessment: Checking Correctness Of Experiments:**

I carefully checked the experiments.

**Review Assessment: Thoroughness In Paper Reading:**

I read the paper at least twice and used my best judgement in assessing the paper.

---

> ### Author Response · Authors · 2019-11-15
> **Improvements of Experiments via Ablation Studies and a Humanoid**
>
> We thank the reviewer for the constructive feedback. We hope we are able to address all concerns accurately.
>
> *** Comparison to State-of-the-art ***
>
> Comment: HiDe uses a top down view of the maze and the x y position of the agent, which certainly is more privileged information. If this is correct, first, this should be discussed more thoroughly and directly in the paper. Second, the experimental setup should be elaborated on: is HIRO or HAC modified to include the same information for the top-level policy? Or can HiDe somehow be extended to not require this information?
>
> HiDe has indeed access to further information as it receives a top-down image, while HAC [1] and HIRO [2] do not get such information. We mention this more explicitly in the updated version of the paper. Moreover, we now provide results that compare our approach against [3], which is follow-up work of HIRO that uses a simplified top-down view image as well (see general comment for the results and discussion).
>
> *** Experiment Section Improvements ***
>
> 1. Comment: Table 1 and Figure 5 seem disconnected. In particular, the numbers reported in Table 1 are clearly not achieved in Figure 5. Is the figure cut off early?
>
> Table 1 and Figure 5 may seem disconnected. For the evaluation shown in Table 1, we selected the checkpoint for each seed where the validation success rate during training was highest. Since these checkpoints are taken at different timestamps, the averaged score of Figure 5 may yield the impression of not achieving these scores. We have added a clearer description of this process in the revised version of the paper and apologize for any possible misunderstandings.
>
> 2. Comment: Furthermore, an additional experiment on a more complicated domain would greatly strengthen the paper. A humanoid agent, for example, seems easy to test for the current method.
>
> We have added a humanoid agent as a proof of concept. We thereby highlight that our method allows training the planning layer with a simplistic agent such as a ball and transferring it to a very complex agent such as the humanoid (contribution 3).
>
> 3. Comment: In my opinion, as the paper currently rests heavily on the results, this section should be further improved.
>
> We try to address this by adding further ablation studies to our experiment sections. More specifically, we compare relative and absolute positions, fixed and learned window sizes for the planning layer, and HiDe with and without interface layer.
>
>
> *** References ***
> [1] Andrew Levy, Robert Platt, and Kate Saenko. Learning Multi-Level Hierarchies with Hindsight. InInternational Conference on Learning Representations, 2019.
> [2] Ofir Nachum, Shixiang Shane Gu, Honglak Lee, and Sergey Levine. Data-efficient Hierarchical Reinforcement Learning. In Advances in Neural Information Processing Systems, pp. 3303–3313,2018.
> [3] Ofir Nachum, Shixiang Gu, Honglak Lee, and Sergey Levine. Near-Optimal Representation Learning for Hierarchical Reinforcement Learning. InInternational Conference on Learning Representations, 2019

---

### Official Review · AnonReviewer3 · 2019-10-22
**Official Blind Review #3**

**Rating:** 3

**Review:**

The submission proposes a novel method for explicit decomposition of hierarchical policies for long-horizon navigation tasks. The approach proposes to separate a policy into 3 modules, high-level planner, intermediate planner and low-level control. The evaluation shows that explicit decomposition is well suited for generalisation across a limited set of RL domains.

The proposed method integrates aspects from a variety of recent work including planning layers from value propagation networks, hindsight training paradigms from hierarchical actor critic and hindsight experience replay and related techniques. The variety of different techniques combined instead of a single main contribution renders it challenging to follow all aspects and in particular to trace relevant contributions to performance - which is rendered harder by a limited evaluation section.

While the approach shows good performance against a couple of start of the art methods, it is necessary to provide sufficient ablations to enable long-term insights for the community. The submission high level goal (explicit decomposition and information asymmetry) is clear, the execution involves the combination of many existing techniques plus variations such that it is hard to make solid statements about the relevance of any part.

It is commendable that the authors have introduced adaptations and improvements to their baselines for a stronger and fairer comparison but the evaluation remains very limited.
I suggest to run different domains as given by other domains from OpenAI gym or DeepMind control suite. But more importantly I suggest to run further ablations without the intermediate planning layer & with absolute goal positions. Furthermore, since HAC seems to perform worse when combined with the proposed low and mid level policies (RelHAC), it would make sense to compare to the proposed high-level policy using low and mid level from HAC instead.

The submission provides an overall interesting perspective but makes it hard to narrow down on contribution and important insights by being unclear in formulation and providing only very limited ablations.

Minor issues include:
- Missing description of the mid level policy - what encourages the proposal of closer short-term goals.
- Missing literature on information asymmetry in RL (e.g. see Tirumala et al 2019 ‘Exploiting Hierarchy for Learning and Transfer in KL-regularized RL’)
- Unclear description of how models that get attached to existing planning layers have been trained (Sec 5.3).
- Additional unclear description in the description of the experiments and method sections (4.2, 4.3)

**Experience Assessment:**

I have published one or two papers in this area.

**Review Assessment: Checking Correctness Of Derivations And Theory:**

I assessed the sensibility of the derivations and theory.

**Review Assessment: Checking Correctness Of Experiments:**

I carefully checked the experiments.

**Review Assessment: Thoroughness In Paper Reading:**

I read the paper thoroughly.

---

> ### Author Response · Authors · 2019-11-15
> **Ablations and Clarifications**
>
> We thank the reviewer for the constructive comments. We have tried to address the reviewer’s concerns about insufficient ablations and clarifications about our contributions.
>
> *** Ablation Studies ***
> Comment: I suggest to run further ablations without the intermediate planning layer & with absolute goal positions
>
> 1. We conduct an ablation study to empirically show that the mid-layer is a necessary part of the architecture (see general comment for the results and discussion).
>
> 2. We provide an ablation study with absolute against relative positions, showing that relative positions are an important aspect of our method in order to both solving the task and achieving generalization to other environments (see general comment for the results and discussion).
>
> *** Additional Domain Experiment***
> Comment: I suggest to run different domains as given by other domains from OpenAI gym.
>
> We have added a humanoid agent as an additional domain for a proof of concept of our method. We thereby highlight that our method allows training the planning layer with a simplistic agent such as a ball and transferring it to a very complex agent such as the humanoid.
>
>
> *** Combining our Planning Layer with original HAC***
> Comment: Since HAC seems to perform worse when combined with the proposed low and mid level policies (RelHAC), it would make sense to compare to the proposed high-level policy using low and mid level from HAC instead.
>
> The low and mid level from original HAC cannot be directly combined with our planning layer, since the goal space in HAC is equal to the state space, such that the subgoals given to layers below consist not only of position goals, but also goals for the proprioceptive state of the agent, such as the joint angles and joint velocities. Contrarily, our planner is decoupled from such a task and therefore only learns to give position goals to the mid level.
>
> *** Minor Comments ***
> 1. Comment: Missing description of the mid level policy - what encourages the proposal of closer short-term goals.
>
> The interface layer is motivated by results from previous work [1]. Furthermore, it is also claimed by [2] that the use of more layers leads to better exploration. To empirically show that the mid-layer is a necessary part in the architecture, we ran an ablation study (see general comment for the results and discussion).
>
> 2. Comment: Missing literature on information asymmetry in RL (e.g. see Tirumala et al 2019 ‘Exploiting Hierarchy for Learning and Transfer in KL-regularized RL’)
>
> We have added the missing literature [3] to the revised version of the paper.
>
> 3. Comment: Unclear description of how models that get attached to existing planning layers have been trained (Sec 5.3).
>
> We have updated and clarified the description of how the transferred models get trained. More specifically, we evaluate the functional decomposition of HiDe by testing the compatibility of layers trained by different agents. We trained HiDe with either the Ball or the Ant agent. Then, we use the planning layer from such an agent and transfer it onto another agent that was trained using RelHAC, i.e., HiDe without our proposed planner on the top layer. The planning layer agent is trained in the more complex environment of experiment 2 and the second agent is trained in the easier environment from experiment 1. Despite being trained in different environments and with different agents, the planner is transferable. Moreover, our estimate indicates that training the planner with a Ball and then transferring it to a more complex agent is as much 3 to 4 times faster than training HiDe with the Ant or Humanoid from scratch.
>
> 4. Comment: Additional unclear description in the description of the experiments and method sections (4.2, 4.3)
>
> Could the reviewer be a bit more specific about what we can clarify/improve in Sections 4.2 and 4.3? We will gladly apply such changes.
>
> *** References ***
> [1] Andrew Levy, Robert Platt, and Kate Saenko. Learning Multi-Level Hierarchies with Hindsight. InInternational Conference on Learning Representations, 2019.
> [2] Anonymous Authors, Why Does Hierarchy (Sometimes) Work So Well in Reinforcement Learning?, under double blind review for ICLR 2020, https://openreview.net/forum?id=rJgSk04tDH
> [3] Dhruva Tirumala, Hyeonwoo Noh, Alexandre Galashov, Leonard Hasenclever, Arun Ahuja, Greg
> Wayne, Razvan Pascanu, Yee Whye Teh, and Nicolas Heess. Exploiting Hierarchy For Learning
> and Transfer in KL-regularized RL. CoRR, abs/1903.07438, 2019

---

### Author Response · Authors · 2019-11-15
**Minor Changes and References**

*** Minor Changes ***
Table 2 and 3 in the original submission contained one wrong entry due to incorrect aggregation. We now updated these tables. The data stems from the _unedited_ Tables 7 and 8 (prev 5 and 6) in supplementary materials.
==================================================
Experiment 2		Ant (before)		Ant (now)
Forward				81+-8			81+-8
Random				90+-2			89+-3
Backward			58+-10			56+-8
Flipped				56+-10			74+-11
==================================================

============================================================
Experiment 3	Ant -> Ball (before)		Ant -> Ball (now)
Forward			100+-0					100+-0
Random			97+-1					97+-1
Backward		90+-22					98+-4
Flipped			81+-44					100+-0
============================================================

*** References ***
[1] Andrew Levy, Robert Platt, and Kate Saenko. Learning Multi-Level Hierarchies with Hindsight. InInternational Conference on Learning Representations, 2019.
[2] Ofir Nachum, Shixiang Shane Gu, Honglak Lee, and Sergey Levine. Data-efficient Hierarchical Reinforcement Learning. In Advances in Neural Information Processing Systems, pp. 3303–3313,2018.
[3] Ofir Nachum, Shixiang Gu, Honglak Lee, and Sergey Levine. Near-Optimal Representation Learning for Hierarchical Reinforcement Learning. InInternational Conference on Learning Representations, 2019
[4] Anonymous Authors, Why Does Hierarchy (Sometimes) Work So Well in Reinforcement Learning?, under submission for ICLR 2020, https://openreview.net/forum?id=rJgSk04tDH
[5] John Schulman, Filip Wolski, Prafulla Dhariwal, Alec Radford, and Oleg Klimov. Proximal policy optimization algorithms, arXiv preprint arXiv:1707.06347, 2017.

---

### Author Response · Authors · 2019-11-15
**General Comment Part 2**

*** Ablation Study Interface Layer ***
We provide an ablation study where we compare our method against a version of HiDe without an interface layer, verifying findings from other work [1,4] that an additional intermediate layer can improve performance. The results for an ant agent in experiment 2 are as follows:
=============================================================================
Experiment 2		Forward		Random		Backward		Flipped
HiDe no interface	10+-5		46+-16		3+-4			0+-0
HiDe				81+-8		89+-3		56+-8			74+-11
=============================================================================

*** Ablation Absolute vs. Relative Positions ***
We conduct an experiment comparing HiDe with absolute versus relative positions. We train two variants of HiDe with absolute positions, one with randomly sampled goals during training (HiDe-AR) and one with a fixed goal position (HiDe-A). For the HiDe and HiDe-R with relative positions, we use the results reported in the initial submission. We evaluate on experiment 1 (simple navigation tasks) and experiment 2 (complex navigation tasks) from our paper. The results look as follows:

=============================================================================
Experiment 1		Forward		Backward	Flipped
HiDe-A				0+-0		0+-0		0+-0
HiDe-AR				95+-1		52+-33		34+-45
HiDe-R				89+-3		61+-14		90+-3
HiDe				85+-6		55+-20		69+-40
=============================================================================
=============================================================================
Experiment 2		Forward		Random		Backward	Flipped
HiDe-A				0+-0		0+-0		0+-0		0+-0
HiDe-AR				0+-0		0+-0		0+-0		0+-0
HiDe				81+-8		89+-3		56+-8		74+-11
=============================================================================

As seen in the results for experiment 1, HiDe-A never manages to solve the task. When allowed random goals during training as in HiDe-AR, it shows a slightly better performance on the forward task than both HiDe and HiDe-R , but does not manage to generalize as well to unseen environments.
In experiment 2, we observe that neither HiDe-A nor HiDe-AR manage to solve any of the environments. This indicates that when scaling to larger environments, relative goal positions can be crucial in learning how to solve a task. We argue that relative positions have the advantage of being reused for similar paths, therefore generalizing within and beyond the training environment, while in the absolute case, the policy has to learn how to generalize to the whole environment map.

Therefore, the results indicate that  i) relative positions improve performance and are an important aspect of our method to achieve generalization to other environments (contribution 2) and ii) random goal position sampling can help agents, but may not be available depending on the environment. Our approach can handle both random and fixed goals.

*** Ablation Study Fixed vs. Learned Window ***
We provide an ablation study, comparing our learned attention window to fixed window sizes. The results indicate the fixed window size can only achieve comparable performance if the size hyperparameter is correctly tuned per agent and environment.

---

### Author Response · Authors · 2019-11-15
**General Comment Part 1**

We would like to thank the reviewers for their feedback and constructive comments. We have prepared an updated version of the paper addressing the main concerns. All of the presented results have been added to the revised paper. Moreover, we reply to each comment and question more specifically down below.

We would like to emphasize the difficulty of the problem that our method HiDe solves. In all our experiments, HiDe learns from a sparse reward, i.e., the environment gives zero reward for reaching the final goal and -1 in all other cases. As shown in [1], such a task is difficult to solve with standard RL algorithms. Furthermore, HiDe does not rely on any modification of the task configuration, such as random start or goal sampling. In such constraint configurations, the tested baseline methods fail. Hence, we train the baselines with randomly sampled goals in our experiments.

We want to clarify our main contributions:
1. A hierarchical architecture that can be trained end-to-end with Reinforcement Learning  from sparse rewards in a setting without any modification of the task configuration via the use of an effective planner and a novel architecture.
2. We show that our method is able to generalize to new environments, while current state-of-the-art methods need to be retrained for each specific environment they are trying to solve.
3. Our decomposed structure, achieved via information asymmetry, allows learning a planner with a very simple agent such as a ball and transferring this planner to a more complex locomotion agent such as an ant or a humanoid, which neither HAC [1] nor HIRO [2] can achieve because of the shared state space across all layers.

The following are the most important changes that have been updated in the revised version of the paper:

*** Comparison to Top-Down View Approach ***
To address the concern of privileged information available to HiDe, we have conducted an additional experiment comparing our methods with [3], an extension of HIRO, which also has access to top-down view images of the environment. We verify that our method shows better generalization performance (contribution 2), even without being provided with random goal sampling and dense rewards during training (contribution 1) as opposed to [3]. The results for experiment 1, which were added to the revised paper, are as follows:
=============================================================================
Experiment 1                Forward    Backward    Flipped
HIRO follow-up            97 +- 5        0 +- 0        0 +- 0
HiDe-R                           89+-3         61+-14        90 +- 3
HiDe                               85+-6        55+-20        69+-40
=============================================================================
As indicated, even with the improved version of HIRO, which uses dense reward, randomly sampled goals and a top-down view of the environment, the method does not generalize beyond the training configuration and environment. This is in agreement with the original paper where the authors claim and show generalization on the flipped environment, but only for the learned goal space representation. Both policies had to be retrained from scratch, essentially requiring retraining for each environment. HiDe generalizes without retraining (contribution 2).

*** Additional Humanoid Domain***
To further highlight contribution 3, we have added an additional domain of a humanoid agent (17 DoF) as a proof of concept, showing that a planning layer from a simple ball agent that requires only short training time can be transferred to a very complex humanoid agent. As can be seen in the video ( https://drive.google.com/file/d/1nOPaIylOP_hLdZy5TirdHE8LPkqAKQZ4/view ), it can successfully solve the provided tasks. We follow standard procedure of training a humanoid [5], which includes using a shaped reward. We train the humanoid in an empty environment. We then use the planning and interface layer from a ball agent and transfer it to the trained humanoid without any modifications. While transfer of a trained HiDe-planner to a humanoid can be achieved, it is currently not possible to train the humanoid with HiDe end-to-end with only sparse rewards. However, the shown transfer demonstrates the benefits of our method.

---

### Decision · Program_Chairs · 2019-12-19

**Decision:**

Reject

**Comment:**

The submission proposes a complex, hierarchical architecture for continuous control RL that combines Hindsight Experience Replay, vision-based planning with privileged information, and low-level control policy learning. The authors demonstrate that the approach can achieve transfer of the different control levels between different bodies in a single environment.

The reviewers were initially all negative, but 2 were persuaded towards weak acceptance by the improvements to the paper and the authors' rebuttal. The discussion focused on remaining limitations: the use of a single maze environment for evaluation, as well as whether the baselines were fair (HAC in particular). After reading the paper, I believe that these limitations are substantial. In particular, this is not a general approach and its relevance is severely limited unless the authors demonstrate that it will work as well in a more general control setting, which is in their future work already.

Thus I recommend rejection at this time.